# Three-Dimensional Analysis of Posterior Mandibular Displacement in Rats

**DOI:** 10.3390/vetsci9030144

**Published:** 2022-03-20

**Authors:** Ioannis Lyros, Efstratios Ferdianakis, Demetrios Halazonetis, Theodoros Lykogeorgos, Antigoni Alexiou, Konstantina-Eleni Alexiou, Maria Georgaki, Emmanouil Vardas, Zafeiroula Yfanti, Apostolos I. Tsolakis

**Affiliations:** 1Department of Orthodontics, School of Dentistry, National and Kapodistrian University of Athens, 11527 Athens, Greece; stratis-fer@hotmail.com (E.F.); dhal@dhal.com (D.H.); alexiou.antigoni@gmail.com (A.A.); apostso@otenet.gr (A.I.T.); 2“Hatzikosta” General Hospital of Messolonghi, 30200 Messolonghi, Greece; theolyk@gmail.com; 3Department of Oral Diagnosis & Radiology, School of Dentistry, National and Kapodistrian University of Athens, 10679 Athens, Greece; kalexiou20@gmail.com (K.-E.A.); zafeiroula86@gmail.com (Z.Y.); 4Department of Oral Medicine & Pathology and Hospital Dentistry, School of Dentistry, National and Kapodistrian University of Athens, 10679 Athens, Greece; mar1georgaki@gmail.com (M.G.); mbardas@gmail.com (E.V.); 5Department of Orthodontics, Case Western Reserve University, Cleveland, OH 44106, USA

**Keywords:** mandibular growth, mandibular posterior displacement, mandibular length, condylar growth, rat, class III malocclusion, orthodontic treatment

## Abstract

Mandibular protrusion and its treatment is challenging for the orthodontist. The aim of the present research was to identify macroscopic changes in the mandible, based on three-dimensional Cone Beam Computed Tomography analysis. Seventy-two male Wistar rats were divided into two equal groups, experimental (group A) and control (group B). Each consisted of three equal subgroups of 12 rats (A1, A2, A3, B1, B2, B3). Full-cast orthodontic intraoral devices were attached to the maxillary incisors of the experimental animals, and effected functional posterior mandibular displacement. Throughout the experimental period, all animals were fed with mashed food. Animals were sacrificed at 30 days (A1, B1), 60 days (A2, B2) and 90 days (A3, B3). At the 60th day of the experiment, the orthodontic devices were removed from the remaining experimental subgroup A3. Measurements revealed significant differences in the anteroposterior dimensions between experimental and control subgroups. However, the observed changes in the vertical dimensions, Condylion/Go’–Menton and the Intercondylar distance proved insignificant. Posterior mandibular displacement of the mandible in growing rats affects the morphology of the mandible and culminates in the development of a smaller mandible at a grown age.

## 1. Introduction

Orthodontists frequently face the challenge of treating class III skeletal malocclusion during the period of growth [1,2,3,4]. Their intervention may comprise functional removable and fixed appliances [5,6], depending on the severity of the disorder and patient characteristics such as age, gender or the ability to cooperate [7]. It would be interesting to assess the osseous alterations effected by an intraoral device causing distal mandibular displacement and, potentially, growth restriction on the molecular or cellular level [8,9,10,11]. Despite the fact that such appliances were introduced early in the 20th century [12], the exact skeletal effect of the loading is still elusive, and it has been the subject of a recent systematic review [13]. Understanding the pathway of bone remodeling may have repercussions on treatment planning and the stability of the intervention.

Mandibular and condylar growth have been repeatedly studied [14,15], and reportedly they are affected by heredity [16,17], hormones [18,19,20], environment [21], dental occlusion [22], and by systemic disease [23]. The mandible makes a major contribution to the shape and function of the dentofacial complex, and thus, it is regarded as a significant determinant of self-esteem [24,25,26].

Lateral cephalometric radiography remains important in orthodontic diagnostics [27], although it is sometimes omitted [28]. However, traditional two-dimensional (2D) imaging suffers from drawbacks, like overlapping structures, magnification, and linear distortion [29,30,31], which may lead to erroneous interpretation [32,33]. The above-mentioned inherent shortcomings may be overcome with the implementation of three-dimensional (3D) imaging, Cone Beam Computed Tomography (CBCT) [34,35,36,37], which is gradually gaining popularity in Orthodontics [38,39,40], in particular due to the reduced radiation dose compared to traditional computed tomography [41,42], despite its higher dose compared to the traditional (2D) lateral radiograph [43,44], a fact that merits prudence when prescribing such an examination in younger patients [45]. Comparisons involving patients and human skulls have demonstrated the reliability [46] and validity of the CBCT in estimating the actual anatomical distance between assigned cephalometric points [47] and similar [48,49,50] or better [51,52,53,54] estimations of cephalometric points in comparison to (2D) lateral cephalometry. 

The rat is likely the most preferred lab animal for conducting experiments on cranial growth despite existing anatomical and physiological differences with humans [55,56]. There is a long history and experience of using rats to study mandibular and condylar growth [57,58,59]. However, growth evaluation has been-based mainly on 2D lateral radiographs [60,61].

In studies with rats, appliances are used to retract the mandible with the aim to inhibit growth and also to enhance additional orthopedic effects [57]. The present original research aspires to identify macroscopic changes in the mandibular bone as appear in CBCT-based three-dimensional analysis.

## 2. Materials and Methods

The study experimental protocol was approved by the Veterinary Directorate and received protocol number 598742/04-10-2019, registered as EL 25 BIO 05, according to Greek national legislation (P.D 56/2013), conforming to European Directive 2010/63/EE and that of the European Council (276/33/20.10.2010) related to the protection of vertebrate animals used in experiments and for other scientific purposes.

### 2.1. Experimental Design

In the present experimental study, seventy-two (72) four-week-old male Wistar rats were used. After their initial four-week breeding in the Hellenic Pasteur Institute, all the animals were transferred and housed at the Laboratory for Experimental Surgery and Surgical Research “N. S. Christeas” at the University School of Medicine in Athens. Standardization following national and European legislation determined cage selection (Tecniplast S.P.A., Italy) and stable centrally ventilated (15 air changes/h) environmental conditions at 55% relative humidity, temperature at 20 °C ± 2 °C, and artificial 12 h span of alternating light–dark cycles. Access to food and water was ad libitum.

The animals were randomly allocated to equal groups, namely groups A (experimental) and B (control), each been divided into three equally sized subgroups featuring twelve rats (A1, A2, A3, B1, B2, B3). The online Random Team Generator tool was used for the grouping. 

Modified orthodontic intraoral devices that have been previously described [62] were placed in the experimental animals and led to posterior mandibular displacement. The full-cast metal orthodontic devices were constructed in the laboratory, following a digital intraoral scanning (TRIOS 3, 3Shape intraoral scanner) of an animal selected at random. The modified guiding appliances were cemented to the maxillary incisors with zinc phosphate cement (Harvard Cement Normal Setting; Harvard Dental International GmbH, 15366 Hoppegarten, Germany) (Figure 1). During the whole experimental period, all animals (experimental and control) were fed with mashed food, produced by blending pellets with water in standardized proportions to achieve a porridge-like consistency.

In total, the experimental period lasted for 90 days. Animals were sacrificed at 30 days (subgroups A1, B1), 60 days (subgroups A2, B2) and 90 days (subgroups A3, B3). At the 60th day of the experiment, orthodontic devices were removed from the subjects still remaining in the experimental subgroup A3. Throughout the entire experimental period, all animals were kept closely monitored for normal growth and development.

### 2.2. Three-Dimensional Analysis

To determine the three-dimensional morphology of the mandible, initial (day 1 of the experiment) and final (day of sacrifice) CBCTs were performed in every rat. The rats were injected intramuscularly for anesthesia with ketamine-xylazine combination at a dosage of 0.2 mL/kg. Once the rats were adequately sedated, they were positioned in the head-resting cushion. All rats were scanned with the same CBCT unit (New Tom VGi, Cefla SC, Imola, Italy) using the same field of view (8 × 8 cm, high-resolution, denture scan) with exposure settings 110 kV. Each scan was performed by an Oral and Dentomaxillofacial Radiologist, who assessed the presence or absence of obvious motion artefacts. In cases of obvious motion artefacts, the scans were performed again and the volumetric data of all scans were exported as Dicom 3 datasets. Three-dimensional reconstruction and analysis were conducted by using Viewbox software (Viewbox© version 4.1.0.10, dHAL Software, Kifissia, Greece). Table 1 and Figure 2 present the detected mandibular anatomic landmarks, while Figure 3a,b show the performed linear measurements.

### 2.3. Statistics

The groups should be kept sufficiently small, for ethical reasons, while reliably detecting potentially statistical results. The number of animals was calculated using power analysis. In addition, the size of the respective samples in the study was finalized after allowing for the low probability that some experimental animals might not cope with the stress of the experimental process.

Subgroups consisting of 12 rats were calculated using standard statistical criteria (a = 0.05, b = 0.10), yielding a power of 90% for detecting a 0.5 mm difference (26.5 vs. 27.0 SD 0.37) for the primary outcome of the study, namely mandibular length (Condylion–I’). Therefore, 72 rats were used, equally divided into experimental and control group.

To calculate the intra-observer and inter-observer errors, double measurements, 4 weeks apart, were made independently by two observers that were blinded to the groups undergoing evaluation. Lin’s concordance correlation coefficient and Bland and Altman analysis were used for the estimation of inter- and intra-observer agreement [63,64]. 

First, seeking to detect any meaningful differences, the dimensional means of the right and left mandibular sides were calculated and were subsequently used for the statistics. Next, differences in measurements related to “group” and “timing (subgroups)” were assessed using linear regression models. Each measurement was regressed on group, timing, and their interaction. When initial measurements were assessed, models were adjusted for initial weight when appropriate. Models for the final measurements were adjusted for the initial ones. Estimated changes from the initial measurements (final minus initial) were also investigated using regression models with group, timing, and their interaction as dependent variables, adjusting for initial weight when appropriate. When normality assumption for the residuals was violated, quantile regression was used. Estimates were adjusted for multiple comparisons, using the Bonferroni method.

Analysis was performed at the α = 5% level of statistical significance (*p*-value < 0.05 indicates a statistically significant result). Data were coded and analyzed using the statistical software Stata ver.14 (Stata Statistical Software: Release 14, College Station, TX, USA: StataCorp LP.).

## 3. Results

Only minor deviation was observed in Lin’s concordance correlation coefficient and values of 0.98 or greater were predominant. These values indicate excellent agreement in measurements. The level of agreement (LoA) of Bland and Altman analysis is displayed in Appendix A. Descriptive statistics with estimated means and standard deviations (SD) are presented in Table 2 and Table 3.

At baseline, only a few differences were evidenced among the subgroups, as expected, due to the sample’s prior randomization. Contrarily, the final measurements revealed noteworthy changes (Table 4, Table 5 and Table 6).

Comparisons between the experimental and the respective control subgroups (A1 vs. B1, A2 vs. B2, A3 vs. B3) revealed a statistically significant restriction of mandibular body length a (Go’–Menton), mandibular body length b (Go–Menton), and mandibular length (Condylion–I’), across all subgroups (*p* < 0.001). The Coronoid–Menton, Condylion–Menton, Condylion–Id, measurements were also found significantly different. Incisal–Id and Incisal–I’ measurements measured significantly different between A1 vs. B1 and A2 vs. B2 subgroups, respectively, albeit differences did not persist after 90 days of experiment (A3 vs. B3 subgroups). The condylion height (Condylion/Go’–Menton), the ramus height (Condylion–Go’) dimensions and the Intercondylar distance did not show significant difference.

Differences within the subgroups in the experimental group (A1, A2, A3) and within the control subgroups (B1, B2, B3) are presented in Table 5. Table 6 presents the mean difference (Final minus Initial) for each subgroup regarding each measurement. It appears that the rate of mandibular growth was smaller in the experimental group in comparison to the control group. The comprehensive regression model regarding major measurements is depicted graphically in Figure 4, Figure 5, Figure 6 and Figure 7. 

## 4. Discussion

The present study investigates in rats an important feature of human face, the mandibular shape, which is a potential determinant of self-esteem [24,65], it affects the individual’s social interaction [66,67] and professional success [68], and is important in the function of the orofacial complex [69,70]. Mandibular development is multifactorial and is regulated by genetic and environmental variables [21,71]. 

Currently, the rat is the most popular animal used in experimental studies involving anatomy and physiology [72,73], despite the existing differences with humans [55,56,74]. Thus, Wistar rats were used in this study with provision of eliminating potential confounding factors related to their characteristics. Therefore, the animals were all male and had no significant differences in measurements that might be connected to the variables of interest. Although the majority of similar studies also selected the rat, some past research has also reported on rabbits [15,62] and even monkeys [75,76]. 

Despite the randomization, some comparisons among subgroups regarding initial measurements were found to be statistically different. This might be attributed to the small size of each subgroup. Twelve rats were sufficient to detect the difference of interest, as was determined after power analysis, but they might not have been adequate to eliminate differences of initial characteristics. However, regression models concerning final measurements were adjusted for initial ones, when appropriate.

Every possible effort was made to breed the animals in a healthy, safe environment, to provide necessary nutrition, and to treat them with dignity. Assessing the rat final weight, there was no difference between A1-B1, contrary to the observations between A2-B2 and A3-B3 (although the device in the A3 subgroup was removed on the 60th experimental day). It could be hypothesized that, initially, the appliance did not seem to have caused any important difference, but subsequently the animals might have faced trouble with feeding. Nevertheless, the rats continued growing, as there were differences among A1, A2, and A3 (Table 5).

Rats live for 3 years, on average. They develop rapidly and their adolescence ends by the end of the second month of ontogenesis. Thus, a rat at 2 months of age (60 days) is considered a young adult. The period of rapid growth allegedly ends by 5 weeks, whereas at the period from 8 to 16 weeks, growth slows down [73,77,78,79,80]. The experimental period of the present study lasted for 90 days, and the rat age in the last subgroup was 120 days. The differences between subgroups A3 and B3 remained statistically significant 30 days following the removal of the device (from A3 subgroup), meaning that the mandible did not exhibit any post-treatment catch-up growth.

Calculations were not performed separately for the left and right mandibular sides. By contrast, the mean values of the contralateral sides were used and are reported. The orthodontic device that was used in the present study was full-cast, intraoral, and was attached to the maxillary incisors. It was introduced by Desai et al. [62], although their research did not report many details on osseous mandibular change, but focused mainly on the temporomandibular joint. It was adopted by Cholasueksa et al. [8] and Farias-Neto et al. [81], whereas Hua et al. [82] and Wang et al. [59] used modified upper/lower devices, unlike Asano [57], Tsolakis [15] and Teramoto et al. [10], who selected extra-oral appliances. Teramoto et al. mention that the magnitude of the traction was excessive, and thus ended up traumatic [10]. 

The mandibular distal displacement effected by the intraoral device that was cemented to the rat maxillary incisors caused a restriction in mandibular length and in mandibular body length. Both Go and Go’ landmarks were identified because the distal outline of the ramus in rodents appears to be particularly more concave related to human anatomy (Figure 3a,b). Asano appears to have faced the same challenge in highlighting similar landmarks [57]. Such developmental restriction is in agreement with the observations of Desai et al. [62] and Cholasueksa et al. [8], and the conclusions of Asano [57], Farias-Neto et al. [81], and Hua et al. [82].

In addition, the oblique osseous and dento-osseous measurements, as depicted by Coronoid–Menton, Condylion–Menton, Condylion–Id, were also found to be significantly different. Interestingly, Hua et al. found their respective oblique measurement, namely the angle between the axis of the condylar process to the mandibular plane, also changing spatial orientation [82]. 

The present study also identified dental alterations, as manifested by the statistically significant differences between experimental and control animals regarding measurements Incisal–Id and Incisal–I’ between A1 vs. B1 and A2 vs. B2 subgroups, respectively, although differences did not persist after 90 days of experiment (A3 vs. B3 subgroups). Dental attrition of the lower incisors in subgroups A1, A2 during the first 60 days of the experiment due to their contact with the device discontinued after debonding the apparatus. Subsequently, lower incisors resumed eruption and so no difference appeared in the dental crown length between experimental and control subgroups.

Interestingly, the vertical component of the mandibular structure as expressed by the condylion height (Condylion/Go’–Menton) and ramus height (Condylion–Go’) did not show significant difference across the experimental and control subgroups inthe final records. This observation agrees with that of Farias-Neto et al., who did not observe any significant difference considering the ramus height [81]. Asano’s conclusions are similar regarding the condylar height, although he also noticed a thickening of the retromolar region and the condylar neck, potentially due to spatial remodeling [57]. This leads to the conclusion that mandibular retrusion might not be expected to cause unwanted side-outcomes affecting facial appearance. Lastly, the Intercondylar distance remained statistically unaffected, in agreement with the conclusions of Farias-Neto et al. [81]. 

To increase accuracy in identifying anatomical landmarks [46,47] and for increased consistency in measuring dimensions [52], CBCT, a 3D reconstruction method, was used in place of ordinary 2D lateral cephalometric radiography [29,32,33,34,38]. To the best of our knowledge, this is the first study to use CBCT for comprehensive cephalometric evaluation in rats, although various digital radiographic techniques have been used [59,81].

The present study was conducted in rodents with respect, and complying with established legislature and regulations. Such experimentation would be off-limits in humans because the interventions might inflict irreversible changes. On the other hand, the existing differences between rodents and humans call for caution when interpreting the results. The present study may be a contribution to evidence-based decision making in orthodontics when treating skeletal Class III malocclusion, and could intensify the call for further research on the long-lasting effects of such interventions aiming to alleviate facial deviations. A randomized controlled trial should be conducted to validate the present research.

## 5. Conclusions

Posterior mandibular displacement in growing rats alters the mandibular morphology and results in the development of a smaller mandible at a grown age. In the rat, it can be concluded that the effects of distal mandibular displacement follow a consistent temporal pattern and are statistically significant. The present study emphasized the long-term stability of the outcomes, revealing that the mandible does not show catch-up growth following treatment.

## Figures and Tables

**Figure 1 vetsci-09-00144-f001:**
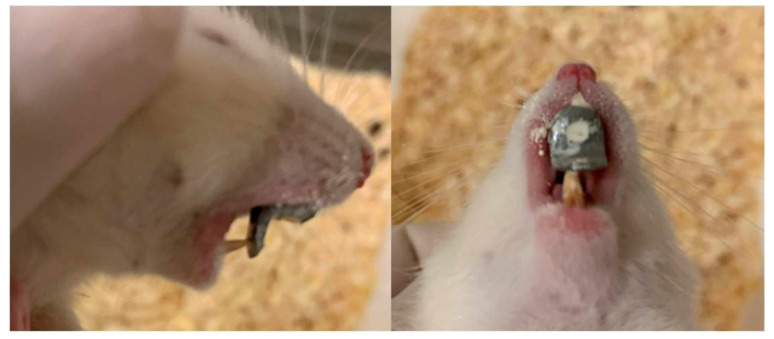
The modified orthodontic intraoral device cemented to the maxillary incisors.

**Figure 2 vetsci-09-00144-f002:**
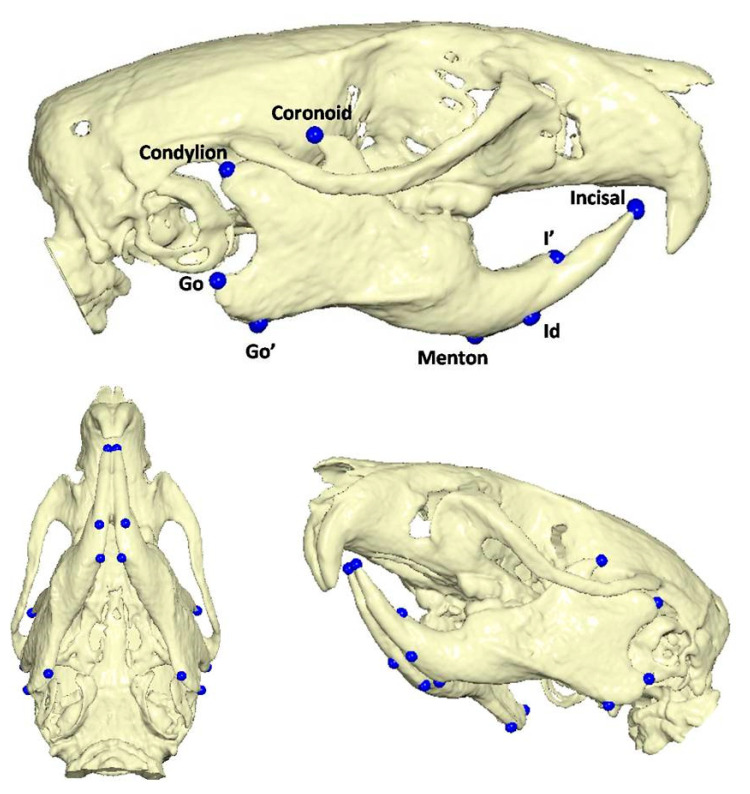
Anatomic landmarks detected in Cone Beam CT reconstructed images.

**Figure 3 vetsci-09-00144-f003:**
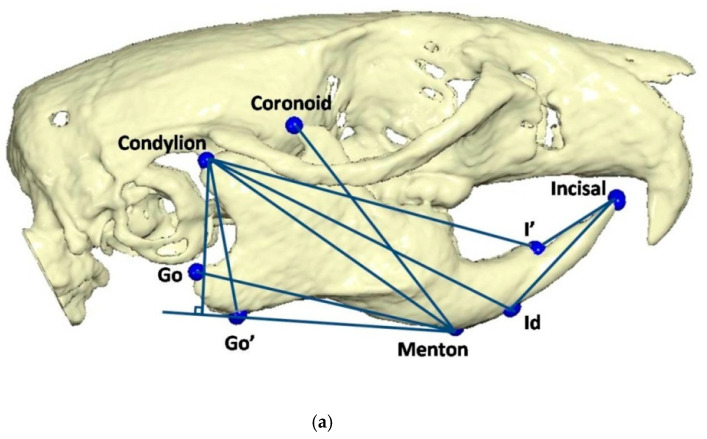
(**a**) Linear measurements: Go’–Menton (mandibular body length a); Go–Menton (mandibular body length b); Coronoid–Menton; Condylion/Go’–Menton (Condylion height); Condylion–Go’ (Ramus height); Condylion–Menton; Condylion–Id; Condylion–I’ (mandibular length); Incisal–Id; Incisal–I’. (**b**) Linear measurement: Condylion right–Condylion left (Intercondylar distance).

**Figure 4 vetsci-09-00144-f004:**
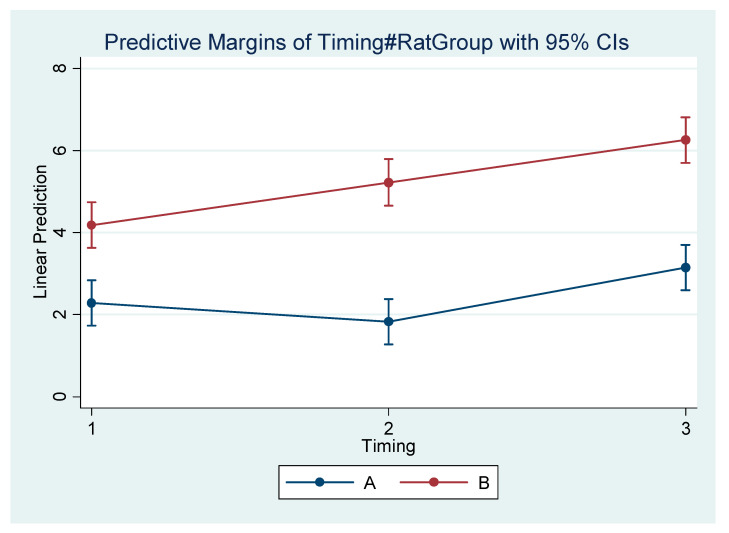
Estimated mean difference (Final–Initial) and 95% Confidence Interval per group and timing in Go’–Menton (mandibular body length a).

**Figure 5 vetsci-09-00144-f005:**
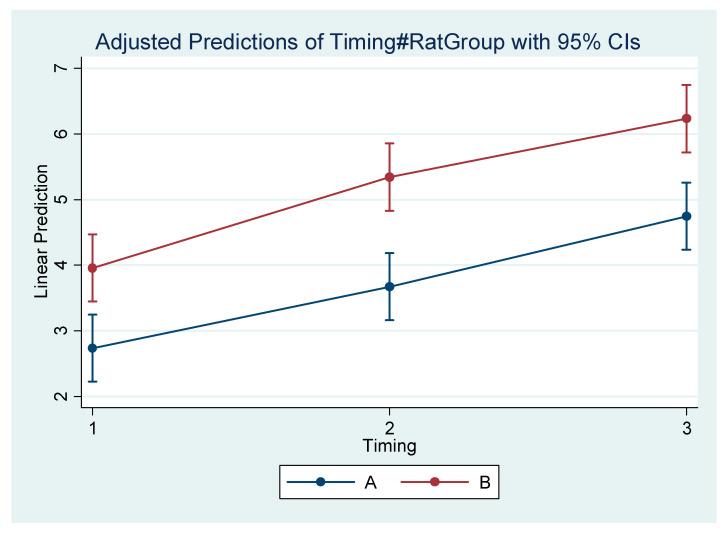
Estimated mean difference (Final–Initial) and 95% Confidence Interval per group and timing in Go–Menton (mandibular body length b).

**Figure 6 vetsci-09-00144-f006:**
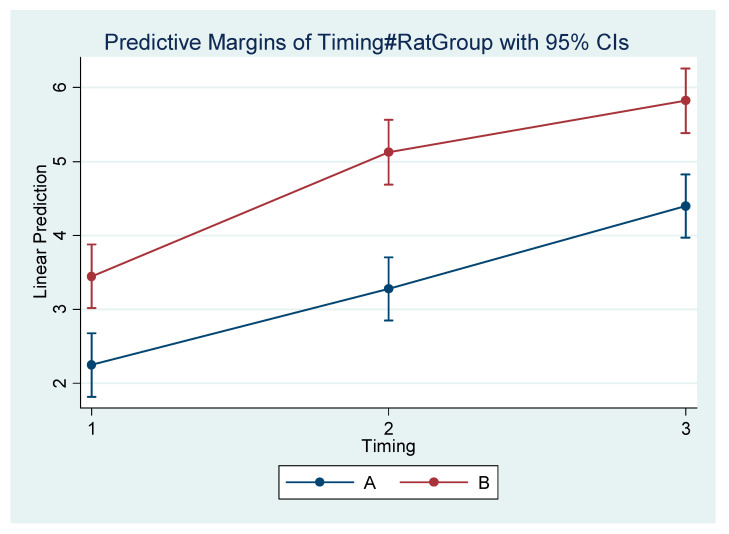
Estimated mean difference (Final–Initial) and 95% Confidence Interval per group and timing in Condylion–I’ (mandibular length).

**Figure 7 vetsci-09-00144-f007:**
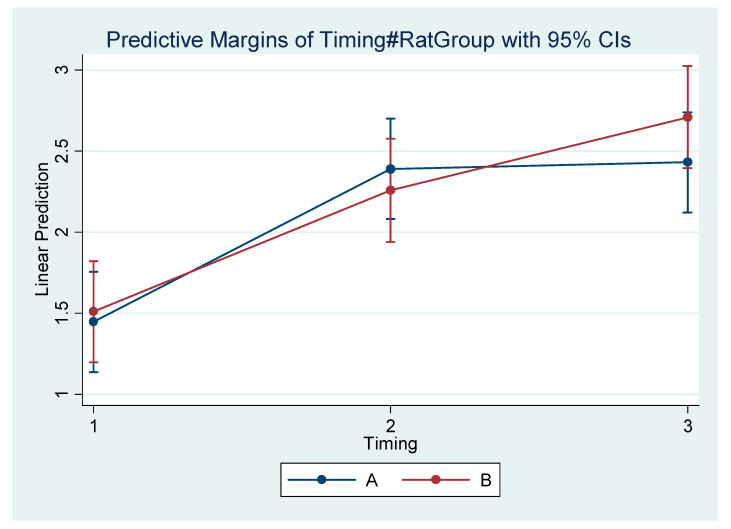
Estimated mean difference (Final–Initial) and 95% Confidence Interval per group and timing in Condylion/Go’–Menton (Condylion height).

**Table 1 vetsci-09-00144-t001:** Description of anatomic landmarks detected in Cone Beam CT reconstructed images.

Anatomic Landmarks	Description
Go’	the lowest point of the gonial process
Go	the most posterior point of the gonial process
Menton	the lowest point of the mental process
Coronoid	the tip of the coronoid process
Condylion	the most posterior and highest point of the condylar process
I’	the most anterior point of the alveolar process at the side of the concavity of the lower incisor
Id	the most anterior point of the alveolar process at the side of the convexity of the lower incisor
Incisal	Incisal edge of the lower incisor

**Table 2 vetsci-09-00144-t002:** Descriptive statistics (mean and standard deviation) for each measurement by subgroups and overall, for experimental group A.

Experimental Group A	Subgroups—Timing				
	A1—0d	A2—0d	A3—0d	Overall	
	Mean (SD)	Mean (SD)	Mean (SD)	Mean (SD)	*p*-Value *
Weight Initial (grams)	117.2 (17.2)	117.6 (17.5)	115.8 (13.4)	116.9 (15.7)	0.956
Go’–Menton Initial (mm)	13.60 (0.52)	14.80 (0.51)	13.70 (0.53)	14.03 (0.74)	**<0.001**
Go–Menton Initial (mm)	16.75 (0.52)	17.14 (0.55)	16.63 (0.46)	16.84 (0.54)	**0.018**
Coronoid–Menton Initial (mm)	16.17 (0.44)	16.48 (0.48)	16.00 (0.56)	16.22 (0.52)	**0.016**
Condylion/Go’–Menton Initial (mm)	8.29 (0.40)	8.20 (0.30)	8.36 (0.33)	8.28 (0.34)	0.137
Condylion–Go’ Initial (mm)	8.81 (0.38)	8.52 (0.31)	8.82 (0.26)	8.71 (0.34)	**0.002**
Condylion–Menton Initial (mm)	18.55 (0.42)	18.91 (0.45)	18.48 (0.42)	18.64 (0.46)	**0.005**
Condylion–Id Initial (mm)	20.65 (0.36)	20.95 (0.52)	20.71 (0.49)	20.77 (0.47)	0.140
Condylion–I’ Initial (mm)	20.85 (0.51)	21.11 (0.48)	20.75 (0.38)	20.90 (0.48)	0.076
Incisal–Id Initial (mm)	7.87 (0.45)	7.94 (0.29)	7.82 (0.28)	7.88 (0.34)	0.741
Incisal–I’ Initial (mm)	5.06 (0.24)	5.13 (0.27)	5.27 (0.26)	5.15 (0.26)	0.124
Intercondylar Initial (mm)	17.78 (0.40)	17.43 (0.33)	17.50 (0.48)	17.57 (0.42)	0.068
	**A1—30d**	**A2—60d**	**A3—90d**	**Overall**	
	**Mean (SD)**	**Mean (SD)**	**Mean (SD)**	**Mean (SD)**	** *p* ** **-Value ***
Weight Final (grams)	256.1 (24.5)	320.7 (25.0)	337.0 (58.6)	304.6 (52.2)	**<0.001**
Go’–Menton Final (mm)	15.86 (0.79)	16.58 (0.69)	16.83 (0.69)	16.43 (0.82)	**0.003**
Go–Menton Final (mm)	19.48 (0.46)	20.81 (0.51)	21.38 (0.77)	20.56 (0.99)	**<0.001**
Coronoid–Menton Final (mm)	18.74 (0.40)	20.13 (0.49)	20.50 (0.98)	19.79 (1.01)	**<0.001**
Condylion/Go’–Menton Final (mm)	9.72 (0.38)	10.58 (0.26)	10.78 (0.51)	10.36 (0.60)	**<0.001**
Condylion–Go’ Final (mm)	10.12 (0.40)	11.02 (0.31)	11.32 (0.53)	10.82 (0.66)	**<0.001**
Condylion–Menton Final (mm)	20.93 (0.48)	22.24 (0.48)	22.87 (0.81)	22.01 (1.01)	**<0.001**
Condylion–Id Final (mm)	23.13 (0.54)	24.48 (0.60)	25.10 (0.95)	24.23 (1.09)	**<0.001**
Condylion–I’ Final (mm)	23.06 (0.48)	24.35 (0.49)	25.13 (0.75)	24.18 (1.04)	**<0.001**
Incisal–Id Final (mm)	7.40 (0.93)	9.22 (1.62)	12.59 (1.00)	9.73 (2.48)	**<0.001**
Incisal–I’ Final (mm)	4.25 (0.97)	5.63 (1.52)	8.90 (1.00)	6.26 (2.29)	**<0.001**
Intercondylar Final (mm)	18.02 (0.35)	18.19 (0.33)	18.08 (0.54)	18.10 (0.41)	0.142

* derived from linear regression models; pairwise comparisons are provided in Table 4 and Table 5.

**Table 3 vetsci-09-00144-t003:** Descriptive statistics (mean and standard deviation) for each measurement by subgroups and overall, for control group B.

Control Group B	Subgroups—Timing				
	B1—0d	B2—0d	B3—0d	Overall	
	Mean (SD)	Mean (SD)	Mean (SD)	Mean (SD)	*p*-Value *
Weight Initial (grams)	109.4 (12.2)	105.4 (22.1)	121.7 (14.7)	112.2 (17.8)	0.062
Go’–Menton Initial (mm)	13.18 (0.43)	13.40 (0.61)	13.74 (0.38)	13.44 (0.52)	0.075
Go–Menton Initial (mm)	16.34 (0.44)	16.28 (0.66)	16.70 (0.55)	16.44 (0.57)	0.804
Coronoid–Menton Initial (mm)	15.90 (0.39)	15.74 (0.74)	15.98 (0.59)	15.87 (0.58)	0.348
Condylion/Go’–Menton Initial (mm)	8.13 (0.34)	8.26 (0.37)	8.55 (0.37)	8.31 (0.39)	**0.040**
Condylion–Go’ Initial (mm)	8.67 (0.34)	8.72 (0.38)	8.97 (0.41)	8.79 (0.39)	0.413
Condylion–Menton Initial (mm)	18.11 (0.50)	18.18 (0.71)	18.50 (0.52)	18.26 (0.59)	0.400
Condylion–Id Initial (mm)	20.36 (0.59)	20.37 (0.81)	20.61 (0.64)	20.45 (0.68)	0.345
Condylion–I’ Initial (mm)	20.42 (0.52)	20.44 (0.78)	20.72 (0.58)	20.53 (0.63)	0.546
Incisal–Id Initial (mm)	7.64 (0.26)	7.73 (0.32)	7.95 (0.54)	7.77 (0.40)	0.381
Incisal–I’ Initial (mm)	5.17 (0.31)	5.15 (0.22)	5.33 (0.43)	5.21 (0.33)	0.865
Intercondylar Initial (mm)	17.60 (0.47)	17.43 (0.50)	17.65 (0.55)	17.56 (0.50)	0.604
	**B1—30d**	**B2—60d**	**B3—90d**	**Overall**	
	**Mean (SD)**	**Mean (SD)**	**Mean (SD)**	**Mean (SD)**	
Weight Final	282.3 (18.1)	365.3 (35.0)	430.2 (30.1)	359.2 (67.4)	**<0.001**
Go’–Menton Final (mm)	17.43 (0.43)	18.74 (0.51)	19.92 (0.80)	18.69 (1.19)	**<0.001**
Go–Menton Final (mm)	20.30 (0.60)	21.63 (0.67)	22.93 (0.56)	21.62 (1.24)	**<0.001**
Coronoid–Menton Final (mm)	19.35 (0.51)	20.63 (0.43)	21.61 (0.47)	20.53 (1.05)	**<0.001**
Condylion/Go’–Menton Final (mm)	9.66 (0.44)	10.56 (0.58)	11.23 (0.23)	10.48 (0.78)	**<0.001**
Condylion–Go’ Final (mm)	9.86 (0.46)	10.72 (0.58)	11.32 (0.25)	10.63 (0.75)	**<0.001**
Condylion–Menton Final (mm)	21.60 (0.67)	23.06 (0.33)	23.91 (0.37)	22.86 (1.08)	**<0.001**
Condylion–Id Final (mm)	23.90 (0.72)	25.79 (0.54)	26.45 (0.41)	25.38 (1.23)	**<0.001**
Condylion–I’ Final (mm)	23.93 (0.58)	25.69 (0.39)	26.44 (0.48)	25.36 (1.17)	**<0.001**
Incisal–Id Final (mm)	10.05 (0.38)	11.06 (0.37)	11.81 (0.48)	10.98 (0.83)	**<0.001**
Incisal–I’ Final (mm)	6.92 (0.23)	7.72 (0.28)	8.13 (0.32)	7.59 (0.58)	**0.001**
Intercondylar Final (mm)	18.02 (0.41)	18.07 (0.48)	18.18 (0.49)	18.09 (0.45)	0.642

* derived from linear regression models; pairwise comparisons are provided in Table 4 and Table 5.

**Table 4 vetsci-09-00144-t004:** Pairwise group per timing measurements comparisons’ *p*-values derived from linear regression models, adjusted for multiple comparison (Bonferroni).

	*p*-Values *		
Measurement	A1 vs. B1	A2 vs. B2	A3 vs. B3
Weight Initial	0.764	0.222	>0.999
Go’–Menton Initial	0.262	**<0.001**	>0.999
Go–Menton Initial	0.436	**0.003**	>0.999
Coronoid–Menton Initial	>0.999	**0.016**	0.810
Condylion/Go’–Menton Initial	>0.999	**0.027**	0.953
Condylion–Go’ Initial	>0.999	**0.001**	>0.999
Condylion–Menton Initial	0.177	**0.005**	0.900
Condylion–Id Initial	>0.999	0.363	0.206
Condylion–I’ Initial	0.345	0.059	0.605
Incisal–Id Initial	0.775	>0.999	>0.999
Incisal–I’ Initial	0.353	0.873	>0.999
Intercondylar Initial	>0.999	0.590	>0.999
Weight Final	0.202	**0.007**	**<0.001**
Go’–Menton Final	**<0.001**	**<0.001**	**<0.001**
Go–Menton Final	**<0.001**	**<0.001**	**<0.001**
Coronoid–Menton Final	**0.002**	**0.001**	**<0.001**
Condylion/Go’–Menton Final	>0.999	>0.999	0.104
Condylion–Go’ Final	0.793	0.056	>0.999
Condylion–Menton Final	**<0.001**	**<0.001**	**<0.001**
Condylion–Id Final	**0.001**	**<0.001**	**<0.001**
Condylion–I’ Final	**<0.001**	**<0.001**	**<0.001**
Incisal–Id Final	**<0.001**	**<0.001**	>0.999
Incisal–I’ Final	**<0.001**	**<0.001**	0.245
Intercondylar Final	>0.999	>0.999	>0.999

* *p*-values in bold indicate statistical significance at 5% level.

**Table 5 vetsci-09-00144-t005:** Pairwise timing per group measurement comparisons’ *p*-values derived from linear regression models, adjusted for multiple comparison (Bonferroni).

	*p*-Values *					
Measurement	A1 vs. A2	A1 vs. A3	A2 vs. A3	B1 vs. B2	B1 vs. B3	B2 vs. B3
Weight Initial	>0.999	>0.999	>0.999	>0.999	0.295	0.075
Go’–Menton Initial	**<0.001**	>0.999	**<0.001**	0.211	0.194	>0.999
Go–Menton Initial	0.116	>0.999	**0.031**	>0.999	>0.999	>0.999
Coronoid–Menton Initial	0.192	>0.999	**0.020**	>0.999	0.627	>0.999
Condylion/Go’–Menton Initial	>0.999	>0.999	0.189	0.153	0.089	>0.999
Condylion–Go’ Initial	**0.016**	>0.999	**0.007**	>0.999	0.959	0.959
Condylion–Menton Initial	**0.027**	>0.999	**0.014**	0.711	>0.999	>0.999
Condylion–Id Initial	0.205	>0.999	0.831	>0.999	>0.999	0.595
Condylion–I’ Initial	0.338	>0.999	0.128	>0.999	>0.999	>0.999
Incisal–Id Initial	>0.999	>0.999	>0.999	>0.999	0.726	>0.999
Incisal–I’ Initial	>0.999	0.192	0.618	>0.999	>0.999	>0.999
Intercondylar Initial	0.099	0.415	>0.999	>0.999	>0.999	>0.999
Weight Final	**<0.001**	**<0.001**	0.999	**<0.001**	**<0.001**	**<0.001**
Go’–Menton Final	>0.999	**0.003**	0.212	0.266	**0.001**	0.152
Go–Menton Final	**<0.001**	**<0.001**	**0.009**	**<0.001**	**<0.001**	**<0.001**
Coronoid–Menton Final	**<0.001**	**<0.001**	**0.023**	**<0.001**	**<0.001**	**0.001**
Condylion/Go’–Menton Final	**<0.001**	**<0.001**	>0.999	**<0.001**	**<0.001**	**0.009**
Condylion–Go’ Final	**<0.001**	**<0.001**	>0.999	**<0.001**	**<0.001**	**0.028**
Condylion–Menton Final	**<0.001**	**<0.001**	**<0.001**	**<0.001**	**<0.001**	**0.005**
Condylion–Id Final	**<0.001**	**<0.001**	**0.012**	**<0.001**	**<0.001**	0.117
Condylion–I’ Final	**<0.001**	**<0.001**	**<0.001**	**<0.001**	**<0.001**	**0.011**
Incisal–Id Final	**<0.001**	**<0.001**	**<0.001**	**0.007**	**<0.001**	0.594
Incisal–I’ Final	**0.003**	**<0.001**	**<0.001**	**0.032**	**0.002**	>0.999
Intercondylar Final	0.198	0.990	>0.999	>0.999	>0.999	>0.999

* *p*-values in bold indicate statistical significance at 5% level.

**Table 6 vetsci-09-00144-t006:** Estimated mean differences (final–initial), 95% Confidence Intervals and *p*-values (compared to 0 i.e., no change) per group and timing.

Final–Initial	Mean Difference (mm)	95% Conf. Interval		*p*-Value
Go’–Menton				
A1	2.29	1.74	2.84	**<0.001**
B1	4.19	3.63	4.74	**<0.001**
A2	1.82	1.27	2.38	**<0.001**
B2	5.23	4.66	5.79	**<0.001**
A3	3.15	2.60	3.70	**<0.001**
B3	6.26	5.70	6.82	**<0.001**
Go–Menton				
A1	2.73	2.22	3.24	**<0.001**
B1	3.96	3.45	4.47	**<0.001**
A2	3.67	3.16	4.18	**<0.001**
B2	5.34	4.83	5.85	**<0.001**
A3	4.75	4.23	5.26	**<0.001**
B3	6.23	5.72	6.74	**<0.001**
Coronoid–Menton				
A1	2.61	2.18	3.03	**<0.001**
B1	3.38	2.96	3.81	**<0.001**
A2	3.69	3.27	4.11	**<0.001**
B2	4.78	4.35	5.21	**<0.001**
A3	4.52	4.10	4.94	**<0.001**
B3	5.73	5.30	6.15	**<0.001**
Condylion/Go’–Menton				
A1	1.45	1.14	1.76	**<0.001**
B1	1.51	1.20	1.82	**<0.001**
A2	2.39	2.08	2.70	**<0.001**
B2	2.26	1.94	2.57	**<0.001**
A3	2.43	2.12	2.74	**<0.001**
B3	2.71	2.39	3.02	**<0.001**
CondylionGo				
A1	1.32	0.98	1.66	**<0.001**
B1	1.17	0.83	1.51	**<0.001**
A2	2.51	2.17	2.85	**<0.001**
B2	1.96	1.61	2.31	**<0.001**
A3	2.51	2.17	2.85	**<0.001**
B3	2.37	2.03	2.72	**<0.001**
Condylion–Menton				
A1	2.42	2.02	2.82	**<0.001**
B1	3.42	3.02	3.82	**<0.001**
A2	3.37	2.97	3.77	**<0.001**
B2	4.77	4.36	5.18	**<0.001**
A3	4.40	4.00	4.80	**<0.001**
B3	5.50	5.10	5.91	**<0.001**
Condylion–Id				
A1	2.51	2.04	2.99	**<0.001**
B1	3.47	2.99	3.95	**<0.001**
A2	3.57	3.10	4.05	**<0.001**
B2	5.29	4.80	5.78	**<0.001**
A3	4.40	3.93	4.88	**<0.001**
B3	5.94	5.45	6.42	**<0.001**
Condylion–I’				
A1	2.25	1.82	2.67	**<0.001**
B1	3.45	3.01	3.88	**<0.001**
A2	3.28	2.85	3.71	**<0.001**
B2	5.12	4.68	5.56	**<0.001**
A3	4.40	3.97	4.82	**<0.001**
B3	5.82	5.39	6.26	**<0.001**
Incisal–Id				
A1	−0.60	−1.37	0.17	0.235
B1	2.40	1.63	3.17	**<0.001**
A2	0.95	0.18	1.72	**0.007**
B2	3.25	2.48	4.02	**<0.001**
A3	4.40	3.63	5.17	**<0.001**
B3	3.85	3.08	4.62	**<0.001**
Incisal–I’				
A1	−0.90	−1.57	−0.23	**0.002**
B1	1.85	1.18	2.52	**<0.001**
A2	0.70	0.03	1.37	**0.033**
B2	2.60	1.93	3.27	**<0.001**
A3	3.60	2.93	4.27	**<0.001**
B3	2.70	2.03	3.37	**<0.001**
Intercondylar				
A1	0.26	−0.09	0.62	0.261
B1	0.37	0.02	0.72	**0.033**
A2	0.79	0.44	1.14	**<0.001**
B2	0.55	0.19	0.91	**0.001**
A3	0.59	0.24	0.94	**<0.001**
B3	0.59	0.23	0.94	**<0.001**

## Data Availability

The data presented in this study are available on request from the corresponding author.

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
