# Peer review of "Three-Dimensional Analysis of Posterior Mandibular Displacement in Rats"

_vetsci, 2022, doi:10.3390/vetsci9030144_

Round 1

Reviewer 1 Report

This is a good animal study regarding the scientific part. The major concern is how important it is for the clinical aspect. The other minor comments are listed as the following:

  1. The usage of some upper- and lower cases seems to be incorrect. The authors should revise the whole manuscript.
  2. For example, the Intercondylar should change to intercondylar (Page 14, Line 311).
  3. The format of the listed references is incorrect for this journal.
  4. Page 3, Line 112: There is an extra space between “light-dark” and “cycles”.

Author Response

  1. The present animal study may be regarded as a significant contribution to clinical practice, because it contributes to the understanding of the mechanism that regulates the mandibular response during functional orthodontic treatment of skeletal Class III malocclusion.
  2. We decided to capitalize the words that indicate the anatomical structures under study and the relative measurements, in order to help the reader follow the experimental process.
  3. We would like to kindly indicate that the manuscript, including the list of references, was prepared in accordance with the journal guidelines.
  4. The extra space in page 3, was tackled.

Reviewer 2 Report

The study that aims to to identify macroscopic changes in the mandibles of rats after the application of full-cast orthodontic intraoral devices is interesting. The use of a three-dimensional Cone Beam Computed Tomography analysis to highlight these changes make the study innovative.

The manuscript is well structure and written, English is adequate.

1)I have a couple of comments:

  • It looks like that the orthodontic intraoral device displaced heavily the mandible both in the anterior-posterior than in the vertical dimensions. Could you please comment on it?
  • In the a priori sample calculation, you use a difference of 0,5 mm for the mandibular length (Condylion-I’). I think that this is a very small difference with no clinical significance at least in human beings. Maybe it is for rats? Could you comment on it?

2) You should check the order of Tables and Figures.

3) I believe that 89 references are a big number for an experimental study, do you think it is possible to reduce it?

Author Response

  1. The device was designed and appropriately fabricated to guide the mandible in posterior occlusion, and thus, there was not any vertical mandibular displacement. That proved highly important for the animals to feed normally.
  2. In rats, the 0.5 mm difference is considered important, because the animal dimensions are much smaller than in humans.
  3. The order of Tables and Figures has been checked and corrected.
  4. We followed your advice and some references considered less than essential were removed.

Reviewer 3 Report

The manuscript “Three-dimensional analysis of posterior mandibular displacement in rats” is a well conducted study. In this study, the authors have identified macroscopic changes in the mandible, based on three-dimensional Cone Beam Computed Tomography analysis. The manuscript needs some major improvements; there are a few suggestions that authors may consider to improve it further:

The use of English language is reasonable, however, there are a number of punctuation and grammatical errors; that should be corrected and rephrased using academic English for a better flow of text for reader.

What is difference between Abstract and Simple Summary?  

-Authors should follow the ARRIVE guidelines for reporting of this manuscript.

-Authors mentioned in discussion that such types of studies in animals cannot be conducted in humans due to probable irreversible damage. Then how will be the results of this study can be further validated?

-Please mention future research directions at the end of the discussion.

-Please carefully check the use of abbreviations throughout the manuscript, figures, tables and abstract.

Results are comprehensive, coving all the results and their discussion is adequate.

There is no presentation of radiographic images and analysis? Please address

Author Response

  1. We would like to inform you that the initial submission of the manuscript was to the journal Animals (MDPI), where the simple summary was required. However, the editorial office suggested the current journal Veterinary Sciences (MDPI) as more appropriate for this type of research and so it was forwarded directly, without revision. Thus, the simple summary has been removed.
  2. The ARRIVE guidelines were used as a checklist in the current manuscript preparation.
  3. When having a new treatment (either pharmaceutical or procedural) researchers first decide to experiment with animals and subsequently organise randomized controlled trials in humans. This initial stage is mandatory in order to avoid irreversible unwanted outcomes occurring to humans.
  4. Future research direction has been added in the discussion.
  5. Abbreviations have been checked.
  6. Radiographic image per se does not contribute to the results. The presented CBCT reconstructed images are of importance for the identification of the anatomic landmarks.

Round 2

Reviewer 3 Report

The authors' revision and response to the comments is satisfactory.

thank you.